# Long-Term Follow-Up of Patients Diagnosed with COVID-19-Associated Pulmonary Aspergillosis (CAPA)

**DOI:** 10.3390/jof8080840

**Published:** 2022-08-11

**Authors:** María Ruiz-Ruigómez, Mario Fernández-Ruiz, Ana Pérez-Ayala, José María Aguado

**Affiliations:** 1Unit of Infectious Diseases, Instituto de Investigación Hospital “12 de Octubre” (imas12), Hospital Universitario “12 de Octubre”, CIBERINFEC, ISCIII, 28041 Madrid, Spain; 2Department of Medicine, Universidad Complutense, Av. Séneca, 2, 28040 Madrid, Spain; 3Department of Microbiology, Instituto de Investigación Hospital “12 de Octubre” (imas12), Universitario “12 de Octubre”, 28041 Madrid, Spain

**Keywords:** COVID-19-associated pulmonary aspergillosis (CAPA), pulmonary aspergillosis, antifungal

## Abstract

COVID-19-associated pulmonary aspergillosis (CAPA) have been documented during the COVID-19 pandemic. The vast majority of these patients do not meet the classic EORTC/MSGERC criteria for invasive pulmonary aspergillosis. The question arises as to whether there may have been an over-diagnosis of this disease. Here we review our experience and analyze the evolution of 27 patients who were diagnosed with CAPA during hospital admission. Surviving patients were followed-up for a mean time of 15 months (SD 3.78) by a group of experts and clinical records of diseased patients were reviewed. After expert evaluation and follow-up, 10 patients were finally assumed as CAPA according to expert opinion. These cases represent 40% of the initially CAPA assumed cases. Our data suggest the need to reconsider actual diagnosis criteria for CAPA what could drive to better identification of these patients.

## 1. Introduction

Since the beginning of the pandemic, there has been concern about the possibility of coronavirus disease 2019 (COVID-19)-related fungal infections in these patients. Based on previous experience of bacterial or fungal infections following influenza infection [1,2] many authors have attempted to anticipate this same scenario in relation to COVID-19.

Among the theories supporting an increased risk of invasive fungal infection (IFI) related to COVID-19 is the direct damage to the airway epithelium caused by the virus, which enables *Aspergillus* to colonize and invade pulmonary tissue, the same as what has been seen with influenza infection [3]. In addition, the fact that the most frequently used treatments for COVID-19, such as steroids and tocilizumab, play an important role in the impairment of immunity supports the possibility of an increased risk of IFI [4] associated with COVID-19.

Reported prevalence of COVID-19-associated pulmonary aspergillosis (CAPA) rates varies widely among the literature (3–34%) [5,6,7,8]. This variability may be due at least in part to the different definitions used, which could have overinflated the reported prevalence of CAPA early in the pandemic [5].

The aim of this study was to briefly review our experience and analyze the long-term evolution of patients who were diagnosed with CAPA in our center in order to assess whether the clinical and radiological evolution was compatible with the diagnosis of invasive aspergillosis.

## 2. Materials and Methods

### 2.1. Study Population and Setting

We performed a retrospective, observational, single-centre study at the University Hospital “12 de Octubre” (Madrid, Spain), a 1250-bed tertiary-care centre. We included all patients admitted with a definite diagnosis of COVID-19 from January 2020 to January 2021 in which *Aspergillus* spp. was isolated in a respiratory sample. Included patients were classified as proven/probable/putative invasive pulmonary aspergillosis or colonization according to both the classification criteria from de European Organization for Research and Treatment of Cancer/Invasive Fungal Infections Cooperative Group and the National Institute of Allergy and Infectious Diseases Mycoses Study Group (EORTC/MSG) [9] and proposed criteria for CAPA [10].

### 2.2. Study Design

Cases were identified through the computerized registry of the Department of Microbiology. Data, including major comorbidities; severity of the underlying disease, hospital admission dates, use of antibiotics, steroids or tocilizumab during hospital admission; need for mechanical ventilation and ICU stay; date of isolation of *Aspergillus* spp., respiratory sample in which *Aspergillus* was isolated, serum galactomannan levels, radiologic test, clinical and radiologic features, treatment, and outcomes, were collected from electronic medical records in a standardized case report form.

### 2.3. Definitions

*COVID-19 confirmed case*: Positive result in the reverse transcription-polymerase chain reaction (RT-PCR) assay for severe acute respiratory syndrome coronavirus 2 (SARS-CoV-2) in nasopharyngeal swab.*CAPA* in immunosuppressed patients: was classified as possible/probable/proven invasive pulmonary aspergillosis according to the EORTC/MSG criteria [9].*CAPA in non-immunocompromised patients*: For these patients we used the 2020 European Confederation of Medical Mycology and International Society for Human and Animal Mycology (ECMM/ISHAM) consensus criteria for the diagnosis of CAPA [10]. According to these criteria, patients with positive SARS-CoV-2 RT-PCR and who develop respiratory insufficiency requiring intensive care could be classified as:
○Proven CAPA: Proven by histopathological and/or direct microscopic detection of fungal elements that are morphologically consistent with *Aspergillus* spp., showing invasive growth into tissues, or by PCR from material that was obtained by a sterile aspiration or biopsy from a pulmonary site, showing an infectious disease.○Probable CAPA: Pulmonary infiltrate or nodules, preferably documented by chest CT, or cavitating infiltrate (not attributed to another cause), or both, combined with mycological evidence: microscopic detection of fungal elements in bronchoalveolar lavage (positive bronchoalveolar lavage culture); positive galactomannan in serum or bronchoalveolar lavage galactomannan; positive aspergillus PCR tests in plasma, serum, whole blood; or bronchoalveolar lavage fluid (modified from ECMM/ISHAM proposed criteria) [10].○Possible CAPA: Pulmonary infiltrate or nodules, preferably documented by chest CT, or cavitating infiltrate (which is not attributed to another cause) in combination with mycological evidence (e.g., microscopy, culture, or galactomannan, alone or in combination) obtained via non-bronchoscopic lavage.*Aspergillus colonization:* Isolation of *Aspergillus* spp. in one or more respiratory samples for which the former criteria was not fulfilled.*CAPA confirmed cases after follow-up:* Surviving patients from our cohort were followed up for at least 6 months after hospital discharge. Two clinicians (MRR and JMA) evaluated individually clinical and radiological evolution of these patients and classified them as true or false CAPA. We considered as CAPA those patients with compatible imaging evolution; cavitating infiltrate or nodules, infiltrates with air-crescent sign, or those with subacute evolution documented by lung CT. We considered as previously misidentified CAPA cases those presenting pulmonary infiltrates or nodules with an accelerated resolution, or those that could be attributed during the follow-up period to another cause.Dose and type of steroids used during study period:
○Metilprednisolone 250 mg/24 h for 3 days;○Metilprednisolone 1 mg/Kg/day for 3–5 days;○Metilprednisolone 40 mg/12 h for 3–5 days.

### 2.4. Statistical Analysis

Quantitative data were shown as the mean ± standard deviation (SD) or the median with interquartile range (IQR). Qualitative variables were expressed as absolute and relative frequencies. Categorical variables were compared using the χ^2^ test. Student’s *t*-test or Mann–Whitney U test were applied for continuous variables, as appropriate.

## 3. Results

From January 2020 to January 2021, *Aspergillus* spp. were recovered from respiratory samples of 279 patients. Of these, 200 were isolated in sputum samples and 79 in lower respiratory tract samples. Twenty-seven patients with *Aspergillus* spp. isolated in a lower respiratory tract sample had a concomitant diagnosis of COVID-19. *Aspergillus* was isolated from tracheal aspirate (BAS) in 25 patients (92.6%), and from bronchoalveolar lavage (BAL) in the remaining two cases.

Demographics and clinical characteristics of these 27 patients are detailed in Table 1. Mean age at diagnosis was 65 ± 9.17 years, with 77.8% (21/27) of the episodes occurring in males. Prior immunosuppression was present in 18.5% (5/27) of the patients, two lung transplant recipients, one allogenic stem cell transplant recipient, one patient with melanoma, and one with an autoimmune disease. Most of the patients (25/27) had previously received broad-spectrum antibiotics. All patients but one were treated with systemic steroids during COVID-19 (96.3% [26/27]), and 40.7% (11/27) received at least one dose of tocilizumab. *Aspergillus fumigatus* was the most common species isolate (44.4% [12/27]) followed by *A. flavus* and *A. niger* (11.1% [3/27]).

Antifungal therapy was initiated in 70.37% (19/27) of episodes; 13/19 (68,42%) received voriconazole; 3/19 (11.1%) liposomal amphotericin B; and 8/19 (42.1%) received isavuconazol. Serum galactomannan assay was performed in 66.7% (18/27), being positive in 11.1% (2/18). Thoracic CT scan was performed in 12/27 episodes and in 5/12 (41.7%) images compatible with IFI were reported. A total of 14/27 (51.9%) deaths occurred.

We analyzed how many patients from our cohort met the ECMM/ISHAM proposed definition criteria for CAPA [10], 25/27 (92,5%) patients met criteria for probable CAPA. Only 5 out of these 25 patients (18.5%) presented prior immunosuppression and thoracic CT patterns compatible with invasive pulmonary aspergillosis, and therefore could be classified as probable cases according to the EORTC/MSG criteria [9]. Thirteen patients with probable CAPA died and therefore no follow-up was possible. Clinical records of these 13 diseased patients were reviewed by experts (MRR and JMA), 10 of them died in the first 10 days of admission: these cases were considered as not CAPA related deaths after expert review. For the remaining three patients death related to CAPA could not be exclude (Figure 1).

In order to analyze the evolution of the surviving patients, they were followed-up in a dedicated COVID outpatient clinic by a group of experts with experience in pulmonary fungal pathology. Twelve surviving patients were followed-up for a mean time of 15 months (SD 3.78); patients clinical, microbiological, and radiological characteristics are represented in Table 2.

We observed that 5 of the 12 survivors (41.6%) were eventually considered as colonization and antifungal treatment was not initiated or was withdrawn within less than 48 h; however, they were followed-up to ensure that they did not subsequently develop symptoms compatible with IFI. Of these five patients, only one had a control CT scan in which no lesions compatible with IFI were observed despite not having received antifungal treatment. The other four patients were followed-up by experts in infectious diseases, none presented respiratory complications during follow-up.

The remaining seven patients were finally assumed as CAPA confirmed cases after follow-up according to expert opinion (Figure 1). Two of these patients had a CT scan in which resolution of pulmonary infiltrates was confirmed. The remaining five patients had at least one chest X-ray performed during the follow-up without IFI compatible images. All seven patients were considered as cured and did not present clinical relapse during follow-up.

## 4. Discussion

Most reported cases of CAPA occurred in patients without classic risk factors of invasive aspergillus infection such as neutropenia or hematological malignancy [11,12,13]; this fact has complicated the diagnosis as patients without histologic evidence of invasive fungal disease cannot meet the classic research definitions of the EORTC/MSG for probable aspergillosis, because these require the presence of immunocompromising host factors [9].

As the pandemic evolves, the question arises as to whether there may have been missed or misidentified individuals. The difficulty of establishing a diagnosis of certainty has been a constant in all series, especially because limited use of diagnostic bronchoscopy due to the need to protect health-care workers from aerosol exposure, and due to the low sensitivity of detection of circulating galactomannan in serum in non-severely immunocompromised patients. Alternative research definitions for CAPA have been proposed [10,14,15], their major limitation may be the characterization of CAPA by nonspecific clinical and radiographic findings that are difficult to distinguish from COVID-19 alone. All these circumstances bring up the concern of the impossibility of distinguishing between colonization and invasive disease.

In the present study we investigated how many patients from the first wave of the pandemic had been initially considered as CAPA and how many of these actually could be considered as CAPA after follow-up. As shown in the flowchart, only 10 patients could be considered as probable CAPA (including 7 survivors and 3 deceased): these cases represent 40% of the initially CAPA assumed cases.

Due to high COVID-related mortality in critical ill patients specially in the first part of the pandemic, we understand the prompt initiation of antifungal treatment in ICU patients for whom respiratory *Aspergillus* spp. has been isolated and fulfil CAPA criteria. Today, with a better knowledge and management of the disease, an increase in the proportion of immunized population and new and promising therapeutic alternatives, it seems necessary to rethink the diagnostic criteria for CAPA.

Our research has some limitations. It is a single-centre study with a small sample size, and there was an absence of tissue-proven diagnosis, which compromised our ability to detect the actual prevalence of CAPA. The strength of our study was the long-term follow-up of the survivors by a group of experts in order to determine whether the clinical picture was compatible with invasive aspergillosis. This long-term follow-up has allowed us to analyze with greater perspective the reality of this problem, concluding that in a considerable proportion of assumed CAPA cases the diagnosis has been overestimated.

## 5. Conclusions

Based on our data and the growing body of published evidence, it seems reasonable to consider the need for stricter criteria that allow a better diagnosis of COVID-19-associated pulmonary aspergillosis.

## Figures and Tables

**Figure 1 jof-08-00840-f001:**
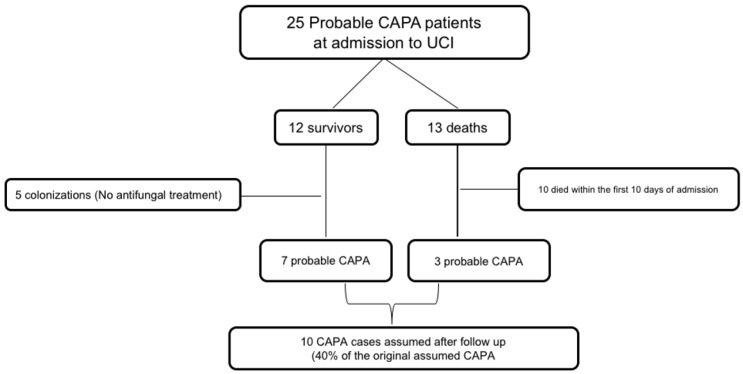
Flowchart showing assumed CAPA patients during admission and CAPA patients confirmed after follow-up.

**Table 1 jof-08-00840-t001:** Demographics and clinical characteristics of the study cohort.

Variable	*n* = 27
Age, years [mean ± SD]	65 ± 9.17
Male gender [*n* (%)]	21 (77.8)
Comorbidity	
Hypertension	12 (44.4)
Diabetes Mellitus	11 (40.7)
Diabetes [*n* (%)]	12 (22.2)
COPD [*n* (%)]	3 (11.1)
Chronic kidney disease [*n* (%)]	4 (14.8)
Prior immunosuppression [*n* (%)]	5 (18.5)
Mechanical ventilation [*n* (%)]	25 (92.6)
COVID-19 Treatment	
Steroids [*n* (%)]	26 (96.3)
Tocilizumab [*n* (%)]	11 (40.7)
Antibiotic therapy [*n* (%)]	25 (92.6)
Isolated *Aspergillus* species, *n* (%)	
*Aspergillus fumigatus*	12 (44.4)
*Aspergillus flavus*	3 (11.1)
*Aspergillus niger*	3 (11.1)
*Aspergillus terreus*	2 (7.4)
*Aspergillus nidulans*	1 (3.7)
*Aspergillus lentulus*	1 (3.7)
*Aspergillus* spp.	2 (7.4)
More than one species in same sample	3 (11.1)
Positive serum galactomannan assay ^a^	2 (11.1)
Compatible IFI images on thoracic CT scan ^b^	5 (41.7)
Antifungal therapy ^c^	
Voriconazole	13 (68.4)
Liposomal amphotericin B	3 (15.7)
Isavuconazole	8 (42.1)
Death	14 (51.9)

COPD: chronic obstructive pulmonary disease; SD: standard deviation. ^a^ Data on serum galactomannan were not available for 9 episodes. ^b^ Data on CT scan images were not available for 15 episodes. ^c^ Data on antifungal therapy were not available for 8 episodes.

**Table 2 jof-08-00840-t002:** Follow-up of patients classified as CAPA (*n* = 12).

Variable	
Age, years [mean ± SD]	65 ± 10
Male gender [*n* (%)]	9 (75)
Follow-up, months [mean ± SD]	15 ± 3.78
**COVID-19 Treatment** [*n* (%)]	
Steroids	12 (100)
Tocilizumab	5 (41.7)
Antibiotic therapy	12 (100)
IFI images on CT scan at CAPA diagnosis ^a^	2 (16.7)
**Isolated *Aspergillus* species [*n* (%)]**	
*Aspergillus fumigatus*	3 (25)
*Aspergillus flavus*	1 (8.3)
*Aspergillus niger*	2 (16.7)
*Aspergillus terreus*	2 (16.7)
*Aspergillus nidulans*	1 (8.3)
*Aspergillus* spp.	1 (8.3)
More than one species in same sample	2 (16.7)
Positive galactomannan in respiratory sample	0 (0)
Positive serum galactomannan assay ^b^	1 (8.3)
**Antifungal therapy [*n* (%)] ^c^**	7 (58.3)
Voriconazole	5 (41.7)
Liposomal amphotericin B ^d^	1 (8.3)
Isavuconazole	2 (16.7)
**CAPA survivor cases assumed after follow-up [*n* (%)] ^e^**	7 (58.3)
Control thoracic CT scan	2 (28.6)
Radiological improvement after treatment	2 (100)
Clinical improvement	7 (100)

^a^ Data on thoracic CT scan images were performed in 5 patients. ^b^ Serum galactomannan was performed in 9 patients. ^c^ 8 of the 13 surviving patients received antifungal treatment. ^d^ Patient treated with amphotericin B received combination therapy with voriconazole. ^e^ After expert follow-up, 7 patients were considered as true CAPA.

## Data Availability

Not applicable.

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
