# Peer review of "Long-Term Follow-Up of Patients Diagnosed with COVID-19-Associated Pulmonary Aspergillosis (CAPA)"

_jof, 2022, doi:10.3390/jof8080840_

Round 1

Reviewer 1 Report

Ruiz-Ruigòmez and colleagues report the results of a retrospective observational study about the follow-up of patients with probable CAPA. According to clinical and radiological findings over a mean follow-up of 15 months, 10 out of 25 probable CAPA cases were assumed to be “real” CAPA cases (7 survived, 3 died).

Data about long term follow-up of patients with CAPA are scarce, thus this study assumes relevance.

I have some notes/questions:

-          Among the 27 patients with Aspergillus spp. isolation, was galactomannan assay performed in respiratory samples? If yes, can you report the positivity rate?

-          A notable finding is that none of the patients who were followed up has a positive galactomannan test on respiratory samples! In previously published studies, galactomannan on respiratory samples (mainly BAL) is the biomarker with the highest positivity rate (i.e. Bartoletti et al, 2021). Can you explain?

-          In Methods, you describe the criteria which lead to establish if the CAPA cases were “true” or “false”, and you mention imaging findings as a criteria. However, among the cases who were followed up, only two of them had control CT scan which showed an improvement. In the other 8 CAPA cases, how did you establish the definitive diagnosis? Please describe.

-          It would be interesting to report patient-level data of the cases assumed to be “real” CAPA (also length of antifungal treatment), you could add a separate table with this information. This would help to understand the process which lead to establish if the CAPA cases were “true” or “false”.

Statistical analysis: comparisons were not made, so line 103-104 are not required.

I think English editing is required before publication.

Author Response

Response to Reviewer 1 Comments

First of all, we would like to thank the reviewer for his/her constructive comments on our research.

Point 1: Among the 27 patients with Aspergillus spp. isolation, was galactomannan assay performed in respiratory samples? If yes, can you report the positivity rate?

Response 1: Unfortunately galactomannan assay was not performed in any of these 27 patients.

Point 2: A notable finding is that none of the patients who were followed up has a positive galactomannan test on respiratory samples! In previously published studies, galactomannan on respiratory samples (mainly BAL) is the biomarker with the highest positivity rate (i.e. Bartoletti et al, 2021). Can you explain?

Response 2: We completely agree on the interest of analyzing galactomannan on respiratory samples. Unfortunately as noted before galactomannan assay in BAL was not performed in our cohort. We have deleted the line in table 2 “Positive galactomannan in respiratory sample” as it can drive to misunderstanding. No galactomannan in BAL was performed so we can´t assume a negative or positive result.

Point 3: In Methods, you describe the criteria which lead to establishing if the CAPA cases were “true” or “false”, and you mention imaging findings as criteria. However, among the cases who were followed up, only two of them had a control CT scan which showed an improvement. In the other 8 CAPA cases, how did you establish the definitive diagnosis? Please describe.

Response 3: We completely agree with the Reviewer on this point. Following the suggestion of Reviewer #1, we have now detailed in the text that even if only 2 of the 7 patients that were followed up had a control CT scan, the other 5 patients had at least one chest x-ray performed that allowed (together with clinical data) experts to assume clinical cure.

Point 4: It would be interesting to report patient-level data of the cases assumed to be “real” CAPA (also length of antifungal treatment), you could add a separate table with this information. This would help to understand the process which leads to establishing if the CAPA cases were “true” or “false”.

Response 4: We appreciate the Reviewer’s comment. Indeed, we acknowledge that there are details on the treatment and analytic changes of patients that are of interest. Even though, we decided not to include this information in detail due to the extension that a brief report allows...

Point 5: Statistical analysis: comparisons were not made, so lines 103-104 are not required.

Response 5: We have removed this point.

Point 6: I think English editing is required before publication.

Response 6: We have make sure to carefully review the language to make sure that it can be easy to understand.

Reviewer 2 Report

The evolution of 27 patients who were diagnosed with CAPA during hospital admission over 15 months.

Of interest, the 27 come from 279 patients [most were presumably without CVOID] with Aspergillus sp. were recovered from sputum. Is this somewhat high for a 12 month period? Was there any exposures in this period [e.g. hospital building work]? Were all of these patients in one ward location – how many were in ICU?

Presumably the period of interest [January 2020 to January 2021] predates the availability of COVID vaccines.

The details of steroid use needs to be detailed. Dose, type and duration.

Of the 13 patients who died – was there any evidence of disease progression. Presumably no post mortem information.

It would be of great interest to compare outcomes with the 252 patients with CAPA but without COVID [are these numbers correct?]. How many of these without COVID met the ECMM/ISHAM proposed definition criteria for CAPA and what was their survival?

Table 2 in the body the footnote ‘e’ needs to be in superscript.

Line 166-167 – I think you are referring here to “..CAPA occurred in patients without classic risk..” in patient positive for COVID.

The overall conclusion appears okay except could be better expressed “Our data suggests the need to reconsider actual diagnosis criteria for CAPA what 20 could drive to better identification of these patients.” Do you mean “Our data suggests a need to refine the diagnostic criteria for CAPA in association with COVID.”

Author Response

Response to Reviewer 2 Comments

First of all, we would like to thank the reviewer for his/her constructive comments on our research.

Point 1: The details of steroid use need to be detailed. Dose, type and duration.

Response 1: Following the suggestion of the Reviewer, we have now detailed in the main manuscript details related to steroid doses used.

Point 2: Of the 13 patients who died – was there any evidence of disease progression. Presumably no post mortem information.

Response 2: Unfortunately we do not have any post-mortem information.

Point 3: It would be of great interest to compare outcomes with the 252 patients with CAPA but without COVID [are these numbers correct?]. How many of these without COVID met the ECMM/ISHAM proposed definition criteria for CAPA and what was their survival?

Response 3: We appreciate the Reviewer’s comment. Indeed, we acknowledge that it could be of great interest to compare patients with COVID-19 Associated Pulmonary Aspergillosis (CAPA) to those with probable IFI and no COVID. Even though we have decided to keep this study as a brief report, this proposed comparison will for sure bring interesting data and could be used to design a score, but unfortunately, this exceeds the aim of our actual study. We will for sure continue to work in this direction.

Point 4: Table 2 in the body the footnote ‘e’ needs to be in superscript.

Response 4: This has been corrected

Point 5: Line 166-167 – I think you are referring here to “..CAPA occurred in patients without classic risk..” in patient positive for COVID.

Response 5: We completely agree with the Reviewer on this point, we are indeed referring to COVID-19-associated pulmonary aspergillosis, what has been called CAPA, and differs from pulmonary IFI (which we refer to when patients have no COVID-19 associated). We hope this point is clear in the text.

Point 6: The overall conclusion appears okay except could be better expressed “Our data suggests the need to reconsider actual diagnosis criteria for CAPA what 20 could drive to better identification of these patients.” Do you mean “Our data suggests a need to refine the diagnostic criteria for CAPA in association with COVID.”

Response 6: Once more we completely agree with the Reviewer on this point. Certainly, our data suggest that a reconsideration of the CAPA criteria may be necessary.

Reviewer 3 Report

The study investigated the clinical significance of Aspergillus isolates in patients with COVID-19. Overall, the study is interesting. I just have several minor comments.

1. Please add one table to summary to casue of excluidng CAPA.

2. Please compare the features of patients with confirmed CAPA vs thos with excluding CAPA.

3. Please revised table 1 to show 25 patients with assumed CAPA as figure 1.

Author Response

Response to Reviewer 3 Comments

First of all, we would like to thank the reviewer for his/her constructive comments on our research.

Point 1: Please add one table to summary to casue of excluidng CAPA.

Response 1: We appreciate the Reviewer’s comment. Indeed, we acknowledge that there are details of patients that are of interest. Even though, we have decided to keep the information of major interest regarding exclusion criteria (mainly not fulfilling defined diagnosis criteria, or presenting a clinical and radiological evolution not compatible with IFI after expert follow-up) to keep this study as a brief report.

Point 2: Please compare the features of patients with confirmed CAPA vs thos with excluding CAPA

Response 2: We appreciate the Reviewer’s comment. Indeed, we acknowledge that it could be of great interest to compare patients with COVID-19 Associated Pulmonary Aspergillosis (CAPA) to those in which CAPA is finally excluded.. Even though we have decided to keep this study as a brief report, this proposed comparison will for sure bring interesting data and could be used to design a score, but unfortunately, this exceeds the aim of our actual study. We will for sure continue to work in this direction.

Point 3: Please revised table 1 to show 25 patients with assumed CAPA as figure 1

Response 3:: We appreciate the Reviewer’s comments. In table 1 we have included information regarding our complete cohort of patients (27 patients with COVID-19 and aspergillosis isolated from low respiratory samples). 25 of these 27 patients developed respiratory insufficiency requiring intensive care which is a major diagnostic criteria of CAPA according to ECMM/ISHAM consensus, so these 25 patients were considered as probable CAPA and therefore had been included in the figure.